# Comparative Analysis of Recent Burn Guidelines Regarding Specific Aspects of Anesthesia and Intensive Care

**DOI:** 10.3390/ebj6040057

**Published:** 2025-10-29

**Authors:** Rolf K. Gigengack, Joeri Slob, J. Seppe H. A. Koopman, Cornelis H. Van der Vlies, Stephan A. Loer

**Affiliations:** 1Department of Trauma and Burn Surgery, Maasstad Hospital, Maasstadweg 21, 3079 DZ Rotterdam, The Netherlands; 2Department of Intensive Care, Amsterdam UMC, De Boelelaan 1117, 1081 HV Amsterdam, The Netherlands; 3Department of Anesthesiology, Amsterdam UMC, De Boelelaan 1117, 1081 HV Amsterdam, The Netherlands; 4Alliance of Dutch Burn Care (ADBC), Burn Center, Maasstad Hospital, Maasstadweg 21, 3079 DZ Rotterdam, The Netherlands; slobj@maasstadziekenhuis.nl; 5Department of Anesthesiology, Maasstad Hospital, Maasstadweg 21, 3079 DZ Rotterdam, The Netherlands; 6Trauma Research Unit Department of Surgery, Erasmus MC, University Medical Centre Rotterdam, Dr. Molewaterplein 40, 3015 GD Rotterdam, The Netherlands

**Keywords:** burn guidelines, airway management, mechanical ventilation, burn resuscitation, pain management, procedural sedation management

## Abstract

Background: Critical care for patients with severe burn injuries is challenging, particularly in the first 24–48 h. Multiple guidelines exist but their recommendations vary in content and in the level of detail. Methods: This narrative review analyzed recent (last 10 years) adult burn guidelines in English, Dutch and German, sourced from PubMed, Medline and official burn society publications. The review focused on airway management, mechanical ventilation, fluid resuscitation, pain management and procedural sedation. Results: All guidelines emphasize early airway assessment and timely intubation in patients at risk for loss of airway patency; however, a strategy for analyzing patients at risk is lacking. Lung-protective ventilation strategy is generally recommended. Fluid resuscitation is the cornerstone during the first phase, though recommendations for thresholds, volume and adjuncts differ. (Chronic) pain management should be multimodal, combining pharmacologic and non-pharmacologic approaches, but specifics on choice of modality are limited, also, there is no uniform strategy for procedural sedation management. Conclusion: Current guidelines offer broadly consistent recommendations for initial burn care but differ in specifics, reflecting evidence gaps. Future guidelines should address advances in airway management, fluid resuscitation endpoints, volume and adjuncts, and give a more detailed (chronic) pain strategy to improve standardization and outcomes.

## 1. Introduction

Critical care of patients with severe burn injury (>15% TBSA) remains challenging and requires special knowledge and experience. In the initial phase (the first 24 to 48 h), maintaining vital functions such as respiration and circulation is crucial. Accordingly, various aspects of airway management, mechanical ventilation, fluid resuscitation and pain treatment play a prominent role in early management [1,2].

Several national and international burn guidelines, including the guidelines of the European Burn Association (EBA), the American Burn Association (ABA), the British Burn Association (BBA), the International Society for Burn Injuries (ISBI) and the German Society for Burn Medicine (DGV) provide recommendations for the management of burn patients [3,4,5,6,7,8,9]. As expected, these guidelines overlap in most recommendations, while differing in others. Recently, Koyro et al. described the similarities and differences between the latest recommendations of six well-known burn societies and analyzed the (dis)advantages of each guideline. They focused on structure, readability, transfer criteria and overall content without comparing the content in detail [10]. Similarly, Paprottka et al. compared the German, European, and American burn guidelines at a more meta-level, comparing readability, completeness, timeliness and overall content without focusing on specific differences in clinical management [11]. To date, no in-depth review has examined the recommendations of the current guidelines for the management of vital functions of adult burn patients in the first 24–48 h. Therefore, this narrative review compares how different guidelines address specific anesthesiological and intensive care aspects during the initial treatment period. Our focus is on airway management and mechanical ventilation, fluid resuscitation, pain management and procedural sedation.

## 2. Materials and Methods

In our narrative review, we considered recent burn treatment guidelines (published within the past 10 years) in English, Dutch or German found through literature searches (PubMed, Medline) or were publicly available from national or international burn societies. We limited our analysis to guidelines that provided recommendations for adult burn patients. Recommendations with respect to the management of vital functions during the initial resuscitation phase of treatment (24–48 h) were extracted. Specifically, recommendations on airway management and mechanical ventilation, fluid resuscitation, pain management and procedural sedation.

## 3. Results

The guidelines of the European Burn Association (EBA) [4], the American Burn Association (ABA) [12], the German Society for Burn Medicine (DGV) [8], the British Burn Association (BBA) [7], the International Society for Burn Injuries (ISBI) [3,9], Japanese Society for Burn Injuries (JSBI) [13], Irish Association of Emergency Medicine (IAEM) [14], Eastern Association for the Surgery of Trauma (EAST) [15], and the Medicines sans Frontieres (MSF) [16] were included in this comparative analysis.

### 3.1. Airway Management and Mechanical Ventilation

A key issue in the acute care of burn trauma patients remains airway management, particularly in patients at risk of airway obstruction following inhalation trauma or direct thermal injury. Table A1 summarizes the recommendations on airway management and Table A2 the recommendations on mechanical ventilation. In this early phase, clinicians must continuously assess and reassess airway patency at adequate intervals and timely secure the airway with an endotracheal tube if patency is at risk (DGV and ISBI recommendations) [8,9]. The decision to intubate is influenced by multiple factors, making it complex and challenging. Endotracheal intubation is an invasive procedure associated with various risks; on the other hand, an endotracheal intubation secures airway patency and eliminates the risk of airway compromise (recommendations of DGV, JSBI and ISBI) [8,9,13]. Scoring systems developed to support this decision still lack sensitivity and specificity. A higher sensitivity often comes at the expense of specificity [17]. The ISBI, DGV, IAEM and EAST guidelines contain specific recommendations for the decision to intubate [8,9,14,15]. They highlight the risk of loss of airway patency due to swelling and mucosal injury from thermal damage. Common indication for intubation includes acute respiratory failure, loss of protective reflexes, airway obstruction, inspiratory stridor and severe cognitive impairment [8,9,14,15]. Burn-specific indication includes systemic inhalation injury (e.g., carbon monoxide poisoning), extensive burns (>40% TBSA) and signs of burns to the oropharynx (e.g., blistering of the mucosa, hoarseness and stridor development) [8,9,14,15]. Careful and continuous airway monitoring is crucial when considering and deciding for deferred intubation [8,9,13,15]. At the same time, when decided to intubate, the physician should assume that burn patients will have a difficult airway. Clinicians must be prepared accordingly and follow the airway management guidelines for difficult airways [18]. The ISBI guideline specifically recommends that the most experienced physician performs the procedure to intubate [9,14].

With respect to mechanical ventilation in burn patients, two guidelines (DGV and ISBI) recommend the use of lung-protective ventilation strategy. The DGV guideline recommends the use of a PEEP-value of ≥5 cm H_2_O, tidal volumes of 6–8 mL/kg, a maximum plateau pressure of 30 cm H_2_O and a maximum driving pressure of 15 cm H_2_O. [8] In contrast, the ISBI guideline provides a more general description of lung-protective ventilation strategy and recommends the use of the lowest possible ventilation pressures and tidal volumes without specifying specific volumes or pressures [9]. To achieve lung-protective ventilation in cases of respiratory failure, the DGV guideline further recommends using permissive hypercapnia and prone positioning and considering veno-venous extracorporeal membrane oxygenation (vv-ECMO) as rescue therapy [8]. The ISBI guideline recommends considering reducing metabolism in burn patients to avoid uncontrolled hypercapnia due to hypermetabolism [9]. Both ISBI and DGV guidelines advise against the use of corticosteroids for inhalation injury, and the ISBI, DGV and JSBI advise against prophylactic antibiotics [3,8,9,13]. The various modes of ventilation are not discussed by either guideline except for the JBSI who discusses the potential of high-frequency percussion ventilation [13]. The DGV-guideline, however, recommends returning to early spontaneous breathing [8].

The DGV, JSBI and ISBI guidelines recommend the use of bronchoscopy to assess the severity of a possible inhalational injury and the DGV and ISBI guideline recommend its use to assess potential causes of mechanical ventilation difficulties, such as the presence of bronchial casts [8,9,13]. At the same time, the DGV-guideline advises against endotracheal intubation to facilitate bronchoscopy and against the use of bronchoscopy to remove sooth from the airways [8].

### 3.2. Fluid Resuscitation

All guidelines recommend early and sufficient fluid resuscitation as crucial measure to prevent and treat burn shock, as summarized in Table A3. The EBA, ABA, DGV and ISBI guidelines recommend initiating fluid resuscitation in burns ≥20% TBSA, while the UK, JSBI, IAEM and MSF guidelines recommends a lower threshold of 15% ≥TBSA [4,7,8,9,12,13,14,16]. The EBA and ISBI guidelines recommend calculating the initial fluid infusion rate based on a total fluid resuscitation volume of 2 to 4 mL/kg/TBSA, the IEAM guideline recommends 4 mL/kg/TBSA, the DGV and JSBI guidelines recommend 2 or 4 mL/kg/TBSA and the ABA and MSF guidelines recommend 2 mL/kg/TBSA [4,8,9,12,13,14,16]. The DGV and ISBI guidelines recommend against the use of fluid bolus to improve urinary output and recommend careful titration of the infusion rate, while the MSF recommends an initial fluid bolus of 20 mL/kg in the first hours [8,9,16]. The EBA, ABA, DGV and ISBI guidelines recommend using isotonic crystalloid solutions and the EBA, DGV, JSBI, MSF and ABA guidelines specify using a balanced solution [4,8,9,12,13,14,16]. Furthermore, due to potential concerns regarding increased oxygen consumption by the liver during lactate metabolism and the risk of rebound alkalosis following lactate administration [8], the DGV advises against the use of Ringer’s lactate [19]. The ABA guideline recommends considering albumin administration between 12 and 24 h after burn trauma as an adjunct to improve urinary output, reduce total resuscitation or as a rescue strategy [12]. In contrast, EBA and DGV recommend using albumin only as a rescue when resuscitation goals are not reached with crystalloids only [4,8,12]. The ABA guidelines advise against the use of fresh frozen plasma during resuscitation outside of research protocols [12]. The JSBI guideline gives a weak recommendation for the use of fresh frozen plasma or hypertonic lactate solution and suggests replacing a portion of the resuscitation fluid with hydroxyethyl starch [13]. This recommendation conflicts with other burn guidelines, general intensive care guidelines and regulatory authorities (e.g., FDA and EMA), all of which advise against its use due to safety concerns [4,8,12,20,21].

None of the guidelines recommend the routine use of norepinephrine as a vasopressor during initial burn resuscitation. The EBA guideline advice using norepinephrine only in cases of life-threatening hypotension despite adequate resuscitation [4]. Similarly, the DGV guideline recommends using noradrenaline and the MSF guidelines recommends dopamine or epinephrine in case of persistent hypotension despite resuscitation with adequate volumes of crystalloids and albumin [8,16]. The ABA guideline does not give a recommendation for the use norepinephrine or vasopressin analogues due to lack of evidence [12].

All guidelines recommend using urinary output as primary endpoint for volume resuscitation and for titrating infusion rate, although they specify slightly different target values and ranges. While the ABA and IAEM guidelines recommend a target range of 0.5–1.0 mL/kg/h, the MSF guideline specifies a target value of 1–2 mL/kg/h, the EBA guideline specifies a target value of 0.5 mL/kg/h and the DGV and ISBI guidelines a target range of 0.3–0.5 mL/kg/h [4,8,9,12,13,14,16]. The ISBI suggests that a lower urinary output (<0.5 mL/kg/h) may be acceptable during the first three hours after severe burn trauma [9]. In addition to urine urinary output as the primary endpoint for volume resuscitation, the EBA recommends combining it with mean arterial pressure (>65 mmHg) and lactate (<2 mmol/L) or base excess [4]. The DGV recommends combining urinary output with lactate (<2 mmol/L) or base excess (>−2), heart rate (<110/min), ITBI (<600–800 mL/m^2^), CI (>2.2–3 L/min·m^2^) or ScvO_2_ (>70%) [8]. The MSF guideline recommends combining it with systolic arterial pressure without specifying a target [16]. The EBA guideline is the only one recommending de-escalation of the resuscitation strategy after the first 24 h when clinically feasible [4].

Regarding advanced hemodynamic monitoring, the ABA guideline recommends against using variables derived from transpulmonary thermodilution (such as cardiac index, intrathoracic blood volume index, global end-diastolic volume index, or extravascular lung water index), as these variables may be associated with over infusion. The guideline gives no recommendations for other dynamic parameters, such as stroke volume variation (SVV) or pulse pressure variation (PPV) [12]. The EBA guideline recommends advanced hemodynamic monitoring only in patients who do not respond adequately to volume resuscitation or in complex situations, such as patients with significant comorbidity or trauma. In addition, the EBA guideline recommends the use of dynamic parameters (SVV, PPV or the passive leg raising maneuver) over static parameters (central venous pressure or wedge pressure (PAOP) [4]. In contrast, the DGV guideline recommends using advanced hemodynamic monitoring to potentially detect fluid overload, as excessive or normal values (e.g., ITBVI) may indicate over infusion. In addition, the DGV guideline recommends using advanced hemodynamic monitoring in patients receiving noradrenaline [8]. The JSBI guideline recommends that transpulmonary thermodilution and arterial pulse contour analysis can be used to titrate infusion rate [13].

### 3.3. Pain Management

Burn pain can be severe and pain management is often complex [22]. Burn pain can be classified into acute, procedural, or chronic pain, as well as nociceptive and neuropathic pain [22]. All guidelines provide recommendations for the treatment of acute or nociceptive pain; they are summarized in Table A4. Recommended analgesics regiments include a combination of paracetamol, NSAIDs or metamizole and an opioid [4,5,8,9]. However, none of the guidelines specify most appropriate opioid for use in burn patients. The ABA guideline recommends selecting opioids based on the patient’s physiological status and the physician’s experience [5]. The DGV guideline recommends using gabapentin or pregabalin, amitriptyline and dexmedetomidine or clonidine as adjuvants in the management of acute pain. The ABA guideline recommends dexmedetomidine or clonidine and (es)ketamine [5], while the ISBI guideline recommends clonidine or dexmedetomidine [9]. Most guidelines recommend using an individualized pain management plan based on a locally available burn pain protocol which includes pain assessment tools such as VAS, CCPOT or BPS [4,5,8,9,13,14,16]. Risk factors for neuropathic pain include advanced age, major burn injuries, long hospital stays, alcohol abuse and smoking [23]. The ABA, ISBI and DGV guidelines recommend the additional use of gabapentin or pregabalin for managing neuropathic pain in these patients [5,8,9] and the JSBI guideline recommends the addition of amitriptyline or carbamazepine [13]. Most guidelines recommend additional non-pharmacological interventions such as hypnosis, distraction, relaxation exercises, cognitive behavioral therapy, virtual reality, hypnotherapy and music therapy [4,5,8,9,13]. However, the guidelines do not specify timing, implementation or the most effective therapy.

### 3.4. Procedural Sedation

The recommendations for procedural sedation are summarized in Table A5. The DGV and ISBI guidelines recommend appropriate analgosedation when burn patients must undergo a procedure. While the ISBI guideline recommends avoiding benzodiazepines, the DGV guideline recommends administering a combination of midazolam and esketamine in patients with hemodynamic instability [8]. Upon completion of the weaning process and in anticipation of extubation, the DGV guideline recommends changing the sedation to propofol. Additionally, the ABA, DGV and ISBI guidelines recommend the use of dexmedetomidine, with the ABA guideline recommending dexmedetomidine as the sedative of first choice. The guidelines recommend the use of a validated sedation score (e.g., RASS or SAS score) to monitor the depth of sedation and to establish daily target scores. The DGV guideline further advises daily wake-up calls, especially when using midazolam.

The ABA recommends the use of opioids for analgosedation. Both the ABA and the ISBI suggest considering the addition of (es)ketamine for sedation during procedures [5,9]. In addition, non-pharmacological therapies play a key role [5,8,9]. The DGV guideline recommends the addition of intravenous lidocaine for procedural sedation [8].

## 4. Discussion

Optimal management of burn patients, particularly in the acute phase, requires a nuanced, multidisciplinary approach. Current international and national guidelines—including those from ISBI, DGV, EAST, ABA, JSBI, IAEM, MSF and EBA—provide essential guidance for clinical decision making. But critical gaps exist in their specificity and applicability, especially for clinicians without specialized burn care expertise.

Airway management remains a complex and dynamic challenge in burn care. Although existing guidelines provide burn-specific intubation triggers, they often lack clear decision-making algorithms. Advanced airway techniques such as nasopharyngoscopy and videolaryngoscopy [24] receive little attention, despite their potential to play a vital role in assessing airway patency, risk for airway obstruction and guiding safe intubation strategies. Future guidelines should incorporate these techniques into flowcharts, as they enable direct visualization of the nasopharynx, epiglottis and vocal cords providing valuable insight in the progression of thermal injury to the oropharynx [24]. Videolaryngoscopy is a useful tool for difficult airways and can be used as the primary intubation strategy, giving its association with higher first-attempt success [25].

With regard to mechanical ventilation modes, the current guidelines recommend elements of the current state of the art in mechanical ventilation for ARDS. There are no recommendations for airway pressure relief ventilation (APRV) or high-frequency percussion ventilation (HPFV), although these are commonly used in the US [26]. To date, there is no evidence that burn patients require different mechanical ventilation strategies than non-burn patients [24], and the most recent consensus statement classified HFPV as inappropriate for burn patients [27]. Future guidelines for burn patients should include a more detailed approach to lung-protective ventilation and also include recommendations on the potential benefits of using APRV and HFPV.

Fluid resuscitation continues to be a cornerstone of early burn care in the first 24 to 48 h after severe burn trauma. While there is consensus among guidelines regarding the use of isotonic crystalloids, there are divergent recommendations on the use of adjuncts such as albumin, vasopressors and advanced hemodynamic monitoring reflecting gaps in evidence. There is also consensus in the guidelines regarding urinary output as endpoint; however, it would be useful to examine whether newer, more sophisticated approaches could complement urinary output, with its limitations as the sole endpoint. Concepts such as fluid reactivity and fluid tolerance could serve as a starting point for recommendations in future guidelines [28,29,30]. In addition, future guidelines should also include recommendations for the integration of multiple hemodynamic parameters to guide volume resuscitation. Furthermore, the role of norepinephrine, which is well established in general critical care, remains insufficiently defined in the context of burn shock [31].

Similarly, the precise role of albumin in burn resuscitation remains under debate. Although recent studies have shown that adjunctive albumin can reduce the total volume of resuscitation [32,33,34], consensus regarding its optimal dose, timing and specific indications in burn patients remain lacking. These aspects are currently being investigated in the multicenter, randomized, controlled ABRUPT-II trial (NCT04356859). Frozen fresh plasma has also shown promising results in animal studies, where it reduced endothelial dysfunction and the total resuscitation volume required in burn shock [35]. However, clinical evidence supporting its efficacy in clinical practice is still lacking [12,34].

The management of acute burn pain has evolved into a multimodal approach, and the guidelines recommend integrating both pharmacologic and non-pharmacologic approaches. In contrast, the treatment of chronic and neuropathic pain remains underrepresented in the guidelines despite its high prevalence and significant impact on patients’ quality of life. Moreover, there is a need for improved guidelines regarding procedural pain, analgesic and sedative drug selection and long-term pain management strategies. Among the reviewed guidelines, only the ISBI guideline addresses this issue and recommends primarily focusing on non-opioid treatments [9].

The care of burn patients is demanding and requires considerable resources in terms of infrastructure and expertise. These requirements can pose significant challenges in resource-limited settings or circumstances, such as armed conflicts or mass casualty incidents [36]. Importantly, such scenarios can also occur in resource-rich countries, underscoring the need for inclusion in future guidelines. Currently, the guidelines lack specific recommendations for the treatment of critical burns in these contexts. Future guidelines should close this gap by including recommendations for burn patients in resource-limited settings.

## 5. Conclusions

In summary, the current guidelines provide useful recommendations for the initial management of vital signs. While most recommendations are consistent across guidelines, some discrepancies reflect gaps in evidence. Future guidelines should include specific recommendations on the use of advanced airway management techniques, fluid resuscitation endpoints and strategies for managing chronic pain.

## Data Availability

The original contributions presented in this study are included in the article Further inquiries can be directed to the corresponding author.

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
