# Peer review of "Comparative Analysis of Recent Burn Guidelines Regarding Specific Aspects of Anesthesia and Intensive Care"

_2673-1991, 2025, doi:10.3390/ebj6040057_

Round 1

Reviewer 1 Report

Comments and Suggestions for Authors

This is a well written review and comparison of international guidelines for the early management of burn injured adults. The authors successfully compare the major burn management guideline showing strengths and weaknesses of each and point out some gaps. I have several critiques. First, high frequency percussive ventilation (HFPV) or volume diffusive respiration (VDR) is used commonly for inhalation injury in the US and internationally. I think it deserves a discussion on whether or not this mode of ventilation should be addressed in guidelines. Secondly, I would elaborate on the use of burn resuscitation endpoints. This is an important topic. Most guidelines still use urine output as the primary metric. With advances in hemodynamic monitoring, should newer more sophisticated devices and biochemical tests be used for this purpose. Thirdly, a discussion of the utility of these guidelines in resource limited environments (combat zones, austere environments, etc) and in disaster/ mass casualty incidents deserve some discussion. Should guidelines be written in such a fashion to address these topics.

Author Response

This is a well written review and comparison of international guidelines for the early management of burn injured adults. The authors successfully compare the major burn management guideline showing strengths and weaknesses of each and point out some gaps. I have several critiques.

Comment 1:

First, high frequency percussive ventilation (HFPV) or volume diffusive respiration (VDR) is used commonly for inhalation injury in the US and internationally. I think it deserves a discussion on whether or not this mode of ventilation should be addressed in guidelines.

Thank you for pointing this out. We are aware that other forms of ventilation are often used for burn patients, but these are rarely mentioned in the guidelines. We have changed the section on ventilation modes in the manuscript as follows:

"With regard to mechanical ventilation modes, the current guidelines recommend elements of the current state of the art in mechanical ventilation for ARDS [23]. There are no recommendations for airway pressure relief ventilation (APRV) or high-frequency percussion ventilation (HPFV), although these are commonly used in the US  [24]. To date, there is no evidence that burn patients require different mechanical ventilation strategies than non-burn patients [21], and the most recent consensus statement classified HFPV as inappropriate for burn patients [23]. Future guidelines for burn patients should include a more detailed approach to lung-protective ventilation and also include recommendations on the potential benefits of using APRV and HFPV.”

Comments 2:

Secondly, I would elaborate on the use of burn resuscitation endpoints. This is an important topic. Most guidelines still use urine output as the primary metric. With advances in hemodynamic monitoring, should newer more sophisticated devices and biochemical tests be used for this purpose.

Thank you for this comment. We agree that urinary output as a primary endpoint in the guidelines could be expanded to reflect advances in hemodynamic monitoring and evolving knowledge about fluid responsiveness and fluid tolerance. We have included the following section in the discussion section of the manuscript.

"Although there is consensus in the guidelines regarding urinary output as endpoint, it would be useful to examine whether newer, more sophisticated approaches could complement urinary output, with its limitations, as the sole endpoint. Concepts such as fluid reactivity and fluid tolerance could serve as a starting point for recommendations in future guidelines [25–27]. In addition, future guidelines should also include recommendations for the integration of multiple hemodynamic parameters to guide volume resuscitation."

Comment 3:

Thirdly, a discussion of the utility of these guidelines in resource limited environments (combat zones, austere environments, etc) and in disaster/ mass casualty incidents deserve some discussion. Should guidelines be written in such a fashion to address these topics.

Yes, we agree that care in resource-poor settings should be included in the key guidelines. We have included the following in the ‘Discussion’ section of the manuscript.

"The care of burn patients is demanding and requires considerable resources in terms of infrastructure and expertise. These requirements can pose significant challenges in resource-limited settings or circumstances, such as armed conflicts or mass casualty incidents [35]. Importantly, such scenarios can also occur in resource-rich countries, underscoring the need for inclusion in future guidelines. Currently, the guidelines lack specific recommendations for the treatment of critical burns in these contexts. Future guidelines should address this gap by including recommendations for the treatment of burn patients in resource-limited settings."

Reviewer 2 Report

Comments and Suggestions for Authors

Thank you for the opportunity to review this comparative analysis of recent burn guidelines. Overall, this is an excellent summary of the recommendations from multiple burn societies and organizations. The authors provide thoughtful commentary, particularly regarding airway evaluation and pain control, though there is less critical discussion of fluid resuscitation. I agree that this manuscript is worthy of publication; however, I have a few comments and questions that may help clarify certain points and strengthen the comparative discussion.

Line 139: The DGV specifically recommends against Lactated Ringer’s because of its lactate content. This warrants further explanation, as it contrasts sharply with most other burn society recommendations. The DGV rationale appears to center on concerns about rebound alkalosis and the hepatic metabolism of lactate; referencing Zander et al., 2005 would be more appropriate here than citation 8 (the ISBI practice guidelines).

Lines 145–147: The JSBI guideline’s suggestion to use hydroxyethyl starch should be addressed in the discussion, as this recommendation conflicts with current guidance from other major organizations (e.g., ABA, ESICM, EBA, DGV) as well as regulatory authorities (FDA/EMA), all of which explicitly advise against HES use due to safety concerns.

Author Response

Thank you for the opportunity to review this comparative analysis of recent burn guidelines. Overall, this is an excellent summary of the recommendations from multiple burn societies and organizations. The authors provide thoughtful commentary, particularly regarding airway evaluation and pain control, though there is less critical discussion of fluid resuscitation. I agree that this manuscript is worthy of publication; however, I have a few comments and questions that may help clarify certain points and strengthen the comparative discussion.

Comment 1:

Line 139: The DGV specifically recommends against Lactated Ringer’s because of its lactate content. This warrants further explanation, as it contrasts sharply with most other burn society recommendations. The DGV rationale appears to center on concerns about rebound alkalosis and the hepatic metabolism of lactate; referencing Zander et al., 2005 would be more appropriate here than citation 8 (the ISBI practice guidelines).

Thank you for your comment. We have added the reference to Zander et al. and changed the wording accordingly:

“Furthermore, due to potential concerns regarding increased oxygen consumption by the liver during lactate metabolism and the risk of rebound alkalosis following lactate administration [19], the DGV advises against the use of Ringer's lactate [8].”

Comment 2:

Lines 145–147: The JSBI guideline’s suggestion to use hydroxyethyl starch should be addressed in the discussion, as this recommendation conflicts with current guidance from other major organizations (e.g., ABA, ESICM, EBA, DGV) as well as regulatory authorities (FDA/EMA), all of which explicitly advise against HES use due to safety concerns.

Thank you for pointing this out. There is indeed a discrepancy between the JSBI guideline and all other guidelines and authorities. We have clarified this by adding the following note to the manuscript:

“This recommendation conflicts with other burn guidelines, general intensive care guidelines and regulatory authorities (e.g. FDA and EMA), all of which advise against its use due to safety concerns.’

Reviewer 3 Report

Comments and Suggestions for Authors

The manuscript “Comparative Analysis of Recent Burn Guidelines Regarding Two Specific Aspects of Anesthesia and Intensive Care” provides a comprehensive and clearly written overview of current recommendations for the management of burn patients during the first 24 hours. The presentation of the protocols is particularly commendable and facilitates understanding for clinicians.

1. Readability of appendix tables: Some of the tables in the appendix extend onto subsequent pages, which makes it difficult to follow and compare the recommendations between different guidelines. It would be helpful if the tables in the appendix could be reformatted or reorganized to improve their clarity and ease of use.

Overall, this is a valuable contribution to the field and, apart from the point raised above, I have no further major comments.

Author Response

The manuscript “Comparative Analysis of Recent Burn Guidelines Regarding Two Specific Aspects of Anesthesia and Intensive Care” provides a comprehensive and clearly written overview of current recommendations for the management of burn patients during the first 24 hours. The presentation of the protocols is particularly commendable and facilitates understanding for clinicians.

Comment 1:

  1. Readability of appendix tables:Some of the tables in the appendix extend onto subsequent pages, which makes it difficult to follow and compare the recommendations between different guidelines. It would be helpful if the tables in the appendix could be reformatted or reorganized to improve their clarity and ease of use.

Overall, this is a valuable contribution to the field and, apart from the point raised above, I have no further major comments.

Yes, we agree and have reformatted the tables in the appendix section.

Round 2

Reviewer 1 Report

Comments and Suggestions for Authors

The edited manuscript is well written and satisfies all of the critiques   from the first the first submission.

Reviewer 2 Report

Comments and Suggestions for Authors

Appropriate revisions

Reviewer 3 Report

Comments and Suggestions for Authors

The author has made the revisions. It can be approved for publication.